# Light-Actuated Liquid Crystal Elastomer Prepared by Projection Display

**DOI:** 10.3390/ma14237245

**Published:** 2021-11-27

**Authors:** Juan Chen, Oluwafemi Isaac Akomolafe, Jinghua Jiang, Chenhui Peng

**Affiliations:** Department of Physics and Materials Science, The University of Memphis, Memphis, TN 38152, USA; jchen12@memphis.edu (J.C.); oluwafemi.akomolafe@memphis.edu (O.I.A.)

**Keywords:** photopatterning, director field, liquid crystal elastomers, light-driven

## Abstract

Soft materials with programmability have been widely used in drug delivery, tissue engineering, artificial muscles, biosensors, and related biomedical engineering applications. Liquid crystal elastomers (LCEs) can easily morph into three-dimensional (3D) shapes by external stimuli such as light, heat, and humidity. In order to program two-dimensional (2D) LCE sheets into desired 3D morphologies, it is critical to precisely control the molecular orientations in LCE. In this work, we propose a simple photopatterning method based on a maskless projection display system to create spatially varying molecular orientations in LCE films. By designing different synchronized rotations of the polarizer and projected images, diverse configurations ranging from individual to 2D lattice of topological defects are fabricated. The proposed technique significantly simplified the photopatterning procedure without using fabricated masks or waveplates. Shape transformations such as a cone and a truncated square pyramid, and functionality mimicking the responsive *Mimosa Pudica* are demonstrated in the fabricated LCE films. The programmable LCE morphing behaviors demonstrated in this work will open opportunities in soft robotics and smart functional devices.

## 1. Introduction

Soft active materials with programmable shape changes have shown applications in drug delivery [1], artificial muscles [2,3], bioelectronics [4,5], and biomedicine [6]. Precise control of the directionality and magnitude of mechanical deformation is vital to program the shape-morphing behaviors. As a type of ordered crosslinked polymer, liquid crystal elastomers (LCEs) have shown remarkable structural transformations by patterning the complex molecular orientations [7]. The change in the molecular ordering of LCE under external stimuli such as light [8], heat [9], or humidity [10] gives rise to a variety of three-dimensional (3D) shape morphs thanks to its anisotropic elasticity [11,12,13]. For example, Ware et al. demonstrated arrays of cones and anti-cones transformed from a flat LCE film by pixelating its two-dimensional (2D) director field [14]. Aharoni et al. morphed the arbitrary shape from 2D to 3D in thin LCE sheets via the inverse-design principle, where the shape is programmed by controlling the molecular orientations of the liquid crystal (LC) monomers [15]. Hence, encoding director orientations in LCE to program the shape-morphing remains non-trivial for developing previously non-achievable mechanical deformations and functionalities.

In order to produce LC with spatially-varying orientations, various techniques have been developed to pattern the 2D director field [16,17]. Rosenblatt’s group created arbitrary arrays of 2D topological defects by using the nano-rubbing of atomic force microscope (AFM) probes [18,19]. The LC director field can also be patterned by using 2D channels micro-fabricated by photolithography [15,20] or two-photon polymerization [21]. Recent advances in 3D printing have also enabled complex 3D shape transformations in LCE, but the alignment has to be realized by optimizing the printing path and material viscosity [22]. As such, both the nano-rubbing and 3D fabrication involve complicated processes and costly equipment, which will prolong the fabrication time and offer limited access for beginners [13]. The most commonly used technique in both industry and academia is the so-called photopatterning technique by using light to control the LC director field [23]. To imprint the directional alignment on the surface that the LC molecules will be interfacing with, a photosensitive azo-dye layer is coated on the surface [24]. By shining linearly polarized light on the azo-dye, the dye molecules will be aligned perpendicularly to the linear polarization direction through isomerization [23]. This alignment of azo-dye will be further used to control the orientation of LC molecules that follow the azo-dye orientations. In order to fabricate complex patterning of LC director field, light patterns with spatially-varying linear polarization are created by using the digital micro-mirror device [25], plasmonic metamasks [26,27], waveplates [28,29], or specially-fabricated photomasks [9,30]. These optical elements are only designed to fabricate specific patterns, and if a new pattern is needed, a new device has to be fabricated, which will increase the production cost. If these optical elements are not used, a laser beam with pixel-by-pixel polarization has to be designed to create pixel-by-pixel orientations, which is time-consuming and costly [14]. Thus, an easily operating photopatterning technique without using fabricated masks or waveplates is highly desired.

In this work, we propose a maskless technique based on a projection display system to pattern complex molecular ordering in LCE film. The system is composed of a projector, a rotational linear polarizer, and focusing lenses. By synchronizing the rotation between the projected images and the polarizer, a variety of patterns with spatially varying director fields are created, which include individual topological defects and a 2D lattice of topological structures. The created patterns can be reconfigured by simply programming the designs of projected images and the rotations of the polarizer. When light-actuated LCE sheets are encoded with pre-designed ordered structures, 3D shape changes such as a cone and a truncated square pyramid are created via light irradiation. Based on the obtained shape-shifting and by cutting the LCE with certain geometries, a light-responsive bioinspired functionality similar to the response of *Mimosa Pudica* to the external stimuli is demonstrated. The demonstrated programmability of LCE shape changes by using the maskless photopatterning technique can be integrated with origami (Japanese art of paper folding) or kirigami (Japanese art of paper cutting) designs in the future, enabling further applications in soft robotics and smart materials.

## 2. Experiment Results

### 2.1. Sample Preparation

Glass slides are washed with detergent in an ultrasonic bath for 8 min, then rinsed with deionized water, acetone, and isopropanol in sequence. To remove the excess liquid, the substrates are baked in an oven at 90 °C for 10–15 min. Afterward, they undergo a five-minute ultraviolet (UV) ozone treatment for deep cleaning. Homeotropic alignments are obtained by spin-coating polyimide PI-1211 on a pre-cleaned substrate at 3000 rpm for 30 s. This substrate is baked on a hot plate at 95 °C for 2 min, followed by one-hour baking at 180 °C.

The photopatterned substrate is prepared by spin-coating a layer of photosensitive azo-dye SD1 (DIC INC, Tokyo, Japan) solution on the surface at 3000 rpm for 30 s. The SD1 solution consists of 0.2 wt% SD1, Figure 1c, and 99.8 wt% *N*, *N*-dimethylformamide (DMF). The coated substrate is then subjected to 15-min baking on the hot stage at 120 °C. The molecules of this azo-dye layer will be oriented perpendicular to the linearly polarized light on the photoalignment layer.

### 2.2. Photopatterned Substrates Preparation

The proposed maskless projection display setup is shown in Figure 2a. The setup is made of a projector (Epson, Suwa, Japan), two convex lenses, a controllable rotating polarizer, and a screen. The projector, with a wavelength of 300–700 nm, has a resolution of 1920 × 1080 pixels and projects predesigned segments. Lens1 (LH-2 lens, Thorlabs, Newton, NJ, United States) and lens2 (convex lens, Thorlabs) are used to get the best focus and size of the projected pattern. A linear polarizer is attached to a motorized rotation stage (PRM1Z8, Thorlabs), connected to a brushed DC servo motor controller (Thorlabs). Thus, the angular speed of the polarizer can be precisely controlled by a LabView program. An SD1 substrate is placed on the screen and irradiated by the projected segments with linearly polarized light. The desired director field will be created on the substrate by the synchronized rotation of the polarizer and projection segments.

A director field is designed as $\hat{n}=\left(n_{x},n_{y}\right)=\left(\cos\theta ,\;\sin\theta \right)$, and $\theta \left(x,y\right)=m\tan^{-1}\frac{y}{x}+\theta_{0}$, where *m* is the topological charge, and $\theta_{0}$ indicates the distortion type. The desired director filed is divided into 36 pieces of 10°-segments evenly, Figure 2b. In order to get the best contrast, the segments are designed as white triangles on a black background, Figure 2b,c. These segments are arranged in 36 PowerPoint slides, respectively, which are further projected in sequence on the SD1 substrate with a time interval. The topological charge is defined as $m=R_{1}/R_{2}$, where $R_{1}$ and $R_{2}$ are the angular speeds of the polarizer and segments, respectively. The starting segment is always set along the *x*-axis, and the 36 segments rotate in a clockwise (CW) sense with a time interval of 10 s. Note that an exposure time of 10 s is enough to imprint the alignment in the SD1 layer. As such, the SD1 substrate is irradiated with rotational segments at an angular speed $R_{2}=$ 10°/10 s. By programming the speed, starting position, and rotation direction of the polarizer, different director fields will be obtained.

For example, when the polarizer axis is aligned along the *y*-axis at the beginning of the process, Figure 2c, and it jogs 5° per 10 s in the CW sense, the angular speed of the polarizer is $R_{1}=$ 5°/10 s. The obtained topological charge will be $m=R_{1}/R_{2}=+1/2$, Figure 2d. The desired pattern with +1/2 defect will be created after a full 2π rotation of segments, Figure 2e,f. For the director field of m=+1 with pure splay $\theta_{0}=0$, Figure 2g, the polarizer is programmed by LabView to start along the *y*-axis and maintain a speed of $R_{1}=$ 10°/10 s in the CW sense. Hence, the obtained topological charge is $m=R_{1}/R_{2}=+1$. After a full rotation of segments, polarizing optical microscope (POM) images of the radial pattern with +1 defect, are shown in Figure 2h,i.

Therefore, by varying the speed, starting position, and rotation directions of the polarizer, the topological charge and distortion type can be well-tuned. A variety of half or integer topological defects can be produced without any fabricated masks or waveplates. The capability to create arbitrary topological defects by the maskless photopatterning technique makes it easier to command the intrinsic molecular orders in LCEs, thus enabling programmable 3D shape transformations from 2D LCE sheets, as shown below.

### 2.3. Light Actuated LCE Film with a Cone

A mixture of 67.5 wt% LC monomer 4-methoxybenzoic acid 4-(6-acryloyloxyhexyloxy) phenyl ester (Synthon, Bitterfeld-Wolfen, Germany), Figure 1a, 26.0 wt% LC crosslinker 1,4-Bis-[4-(3-acryloyloxypropyloxy) benzoyloxy]-2-methylbenzene (Wilshire INC, Princeton, NJ, United States), Figure 1b, 5.2 wt% azobenzene photoswitch Disperse Red 1 acrylate (DR1A, Sigma-Aldrich, St. Louis, MO, United States), Figure 1e, and 1.3 wt% photoinitiator (Irgacure 651, Ciba, Basel, Switzerland), Figure 1d, are dissolved in Dichloromethane (DCME, Sigma-Aldrich). Afterward, the solution is placed in the oven at 90 °C overnight to evaporate the solvent.

A cell is constructed by using 20 µm spacers to sandwich a substrate with photoaligned patterns and another substrate with homeotropic alignment. Both alignment surfaces are facing each other to induce a hybrid configuration across the sample, Figure 3a. The prepared LCE monomer mixture is infiltrated into the chamber at 80 °C on a hot stage. The temperature is decreased to 45 °C at the rate of 5 °C/min. The solution is observed to be isotropic at 80 °C and be in the nematic phase at 45 °C. Next, the cell is irradiated by UV light with 1.4 mW/cm^2^ intensity at a wavelength of 365 nm for 15 min to finish the polymerization. The irradiation of the UV light will not influence the photostability of the photopatterned surface, and the molecular orientations will be fixed in the LCE films after polymerization. The cell is split by a razor blade. After sonicating in a petri dish with water for 20 s, the free-standing LCE film can be detached from the substrate by a tweezer.

Figure 3b shows a topological defect of m=+1 with $\theta_{0}=\pi /2$, named circular +1 defect. It is fabricated by starting the polarizer along the *x*-axis and rotating in the CW direction at the same speed of the segments, Figure 3c,d. Under the irradiation of a collimated LED (455 nm, Thorlabs) with light intensity is 690 mW at its 100% beam power, the actuation of the fabricated LCE film with the bottom surface of the designed circular pattern is manifested as a cone, Figure 3e. The light actuation is enabled by the azobenzene photoswitch of DR1A, which changes the LC orders via photoisomerization. The decrease in the nematic ordering in the LCE induces contraction along the alignment direction and expansion perpendicular to it. For the sample in Figure 3e, due to the inside hybrid configuration, the perimeter contracts on the bottom surface, but the radii extend on the top surface, which can be reconciled only if a cone is formed. Moreover, this morphing is reversible, as the amplitude can be actuated by the switching state of light, Figure 3f.

### 2.4. The Morphing of LCE Film with a Truncated Pyramid Shape

In addition to a single cone, a complex 2×2 cone array is enabled by embedding the circular +1 array into the LCE film, Figure 4. As depicted in Figure 4a, the director field is made of four individual circular +1 topological defects without any boundary. To achieve this pattern, the PowerPoint slides are modified to include four identical segments on every single slide, Figure 4b. The distance intervals of the segments along the *x*-axis and *y*-axis are the same. The polarizer and segments all start along the *x*-axis and rotate in the CW direction at the same speed. These four individual defects can be created at the same time after 2π rotation of the exposed segments, Figure 4c. Instead of four independent cones, the overall shape morphing turns out to be in a truncated square pyramid. The actuated amplitude is about 6 mm, which is about 300 times taller than that of a flat sheet with 20 µm thickness, Figure 4d,e. The shape of a truncated pyramid is enabled thanks to the overlapping of these topological defects.

Previously, the shape-morphing of LCE film with a 2D lattice of +1 topological defects was demonstrated by the White group [14,31]. The spatially varying director field was fabricated by shining pixel-by-pixel linearly polarized light, which is time-consuming and costly. In our work, multiple topological defects can be created at the same time, and it only takes six-minute irradiation to complete the whole pattern, which is much more efficient. Meanwhile, by adjusting the positions of the lens or replacing one of them with an objective lens in the setup, scalability of sizes from tens of micrometers to centimeters can also be enabled.

### 2.5. Biomimetic Actuation of LCE Device

The actuation of hierarchical assembly of LCE geometries inspired from the responsive *Mimosa Pudica* to the external stimuli is also demonstrated. *Mimosa Pudica* is a special plant that will close its leaves upon mechanical stimulation, Figure 5a. Here, we fabricated a bioinspired LCE device of *Mimosa Pudica* by using the radial +1 defect with a geometric triangular shape. LCE film with radial +1 defect (shown in Figure 1g) is fabricated and cut into triangular geometries. Then four of them are hierarchically stuck to a paper clip, Figure 5b. Each of the LCE films adopts a hybrid configuration with the top surface of aligned pattern, inset of Figure 5b. Before irradiation, the biomimetic “leaves” hang down owing to a self-weight of 2 mg. Figure 5c–f are snapshots extracted from Appendix A, depicting the actuation of every pair of “leaves” under light irradiation, respectively. Due to the hybrid configuration in the LCE “leaves”, the “leaves” bend upward to the “closed” state via actuation. Similar to *Mimosa Pudica* responding to the external stimulus, the biomimetic “leaves” only morph when the stimulus “light” is present. After “leaves” move away from the stimulus “light”, their shape deformation will be reversed to the original state. The bending angle of the actuator is dependent on the light intensity, Figure 5g. The inset images in Figure 5g are all taken after 5 s of irradiation for different intensities.

## 3. Conclusions

There have been extensive theoretical analyses about mechanical responses of LCE under external stimuli [7,32,33,34,35,36,37,38,39,40]. In this work, the actuation mechanism is rooted in the nematic ordering change in the LCE film. The decrease in the nematic ordering in the LCE induces contraction along the alignment direction and expansion perpendicular to it. For the LCE film combining a top surface of homeotropic alignment with a bottom surface of photoaligned circular pattern, the deformations in the LCE film can only be reconciled by bulging to a cone thanks to this hybrid configuration. Likewise, the shape morphology of a truncated square pyramid is also enabled based on this fundamental understanding.

In summary, we have demonstrated a cheap, efficient, and easy photopatterning technique to create spatially varying molecular orientations in LCE films by using a maskless projection display setup. A variety of individual topological defects and a 2D lattice of defect array are created by this simple method. We further integrate the topological defects into the light-fueled LCE films to achieve programmable, reversible, and remotely-actuated shape changes. The actuation of a cone, a truncated square pyramid, and a biomimetic LCE *Mimosa Pudica* device is realized by light irradiation. The deformation angle is dependent on the light intensity. The demonstrated programmable shape changes of LCE films can be further used in soft robotics, smart functional materials, and biomedical devices.

## Figures and Tables

**Figure 1 materials-14-07245-f001:**
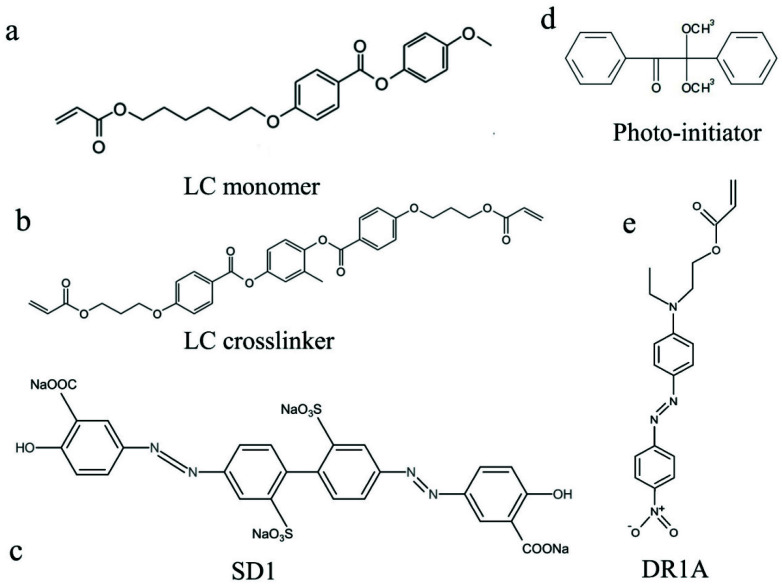
Chemical structures of azo dye and components of LCE. (**a**) LC monomer; (**b**)LC crosslinker; (**c**) Azo dye SD1; (**d**) Photo-initiator; (**e**) Disperse Red 1 acrylate (DR1A).

**Figure 2 materials-14-07245-f002:**
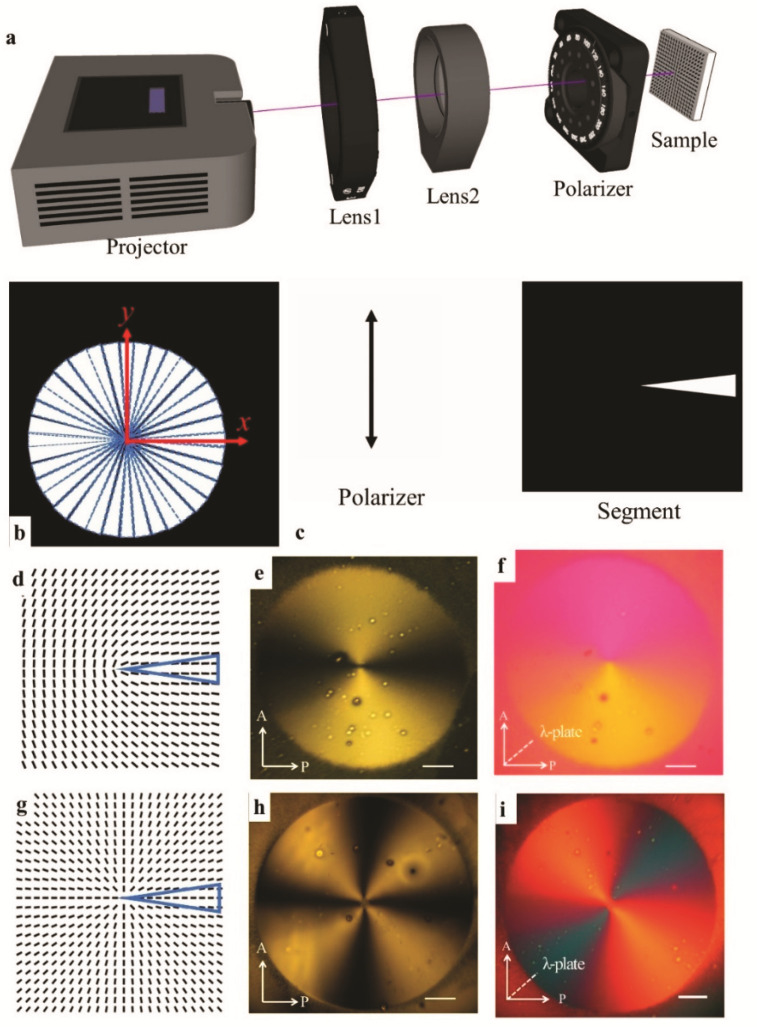
Illustration of photopatterning process. (**a**) A maskless photopatterning setup by projection display. (**b**) The pattern is divided into 36 segments evenly. (**c**) The starting positions of the polarizer and segments for producing director filed in (**d**,**g**). (**d**) The director field of +1/2 defect. (**e**,**f**) The POM images of +1/2 defect without (**e**) and with (**f**) the red plate. If the slow axis of the “red plate” is in the same direction as the LC alignment, the interference color will be “violet” or “blue”, indicating an increase in retardance. (**g**) The director field for a radial +1 defect. (**h**,**i**) The POM images of the radial +1 defect without (**h**) and with (**i**) the red plate. Scale bars in all images are 2.5 mm. The blue triangles in (**d**,**g**) are the starting positions of the segments. P and A represent polarizer and analyzer.

**Figure 3 materials-14-07245-f003:**
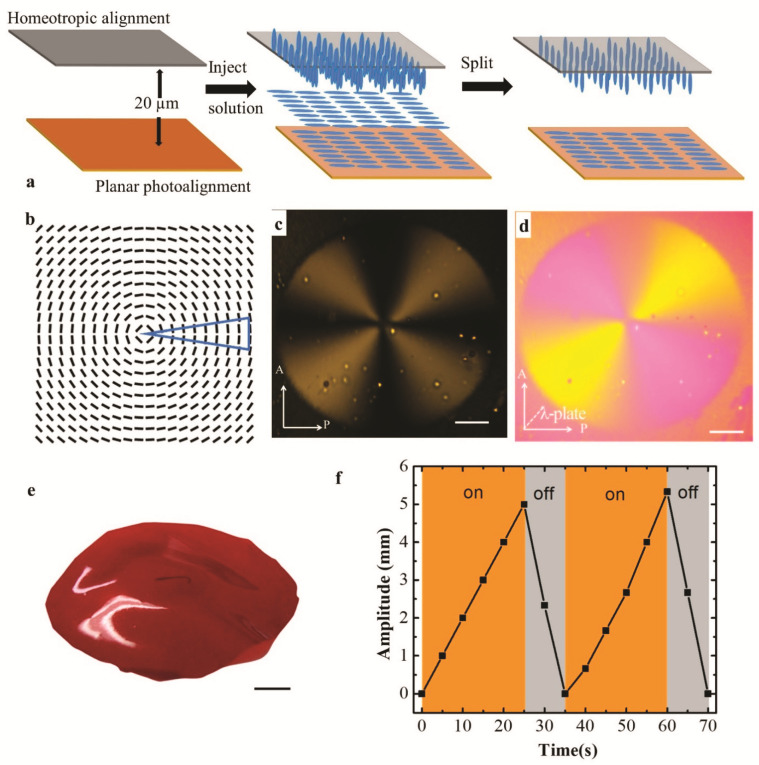
Fabrication of LCE film with circular +1 defect. (**a**) The LCE film fabrication process. (**b**) The director field of circular +1 defect. The blue triangle is the position of the starting segment. (**c**,**d**) The POM images without (**c**) and with (**d**) red plate. (**e**) A single cone is formed in LCE film with the bottom surface of a circular +1 defect. (**f**) The dependence of the amplitude of the LCE film on light irradiation. The scale bars are 2.5 mm.

**Figure 4 materials-14-07245-f004:**
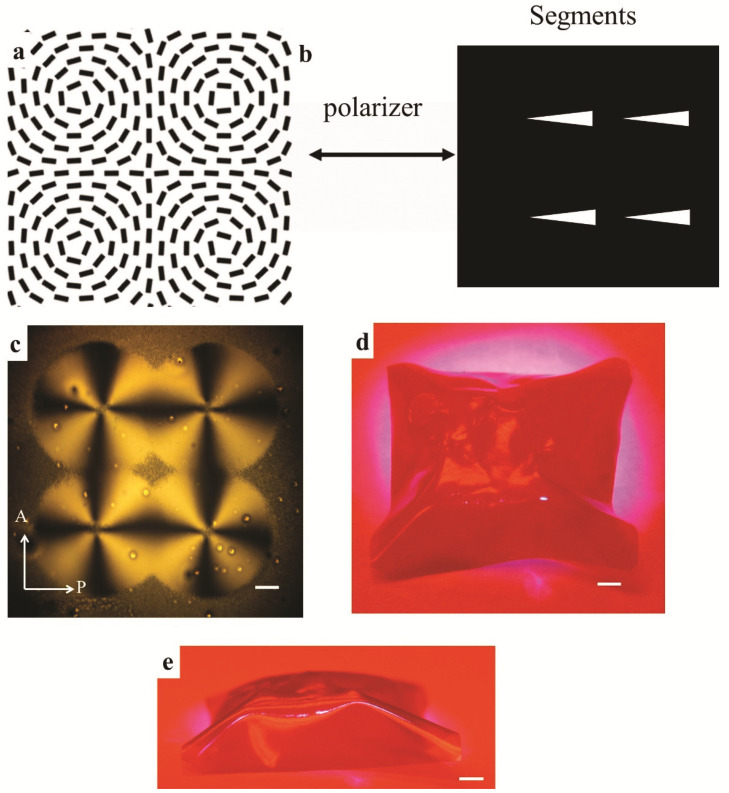
Shape changes of the LCE film with a 2D lattice of circular +1 defects. (**a**) The director filed with a 2D array of circular +1 defects. (**b**) The starting positions of the polarizer and segments. (**c**) The POM image of the designed pattern. (**d**,**e**) The top (**d**) and side (**e**) views of the shaping deformation of the LCE film, respectively. The scale bars are 2.5 mm.

**Figure 5 materials-14-07245-f005:**
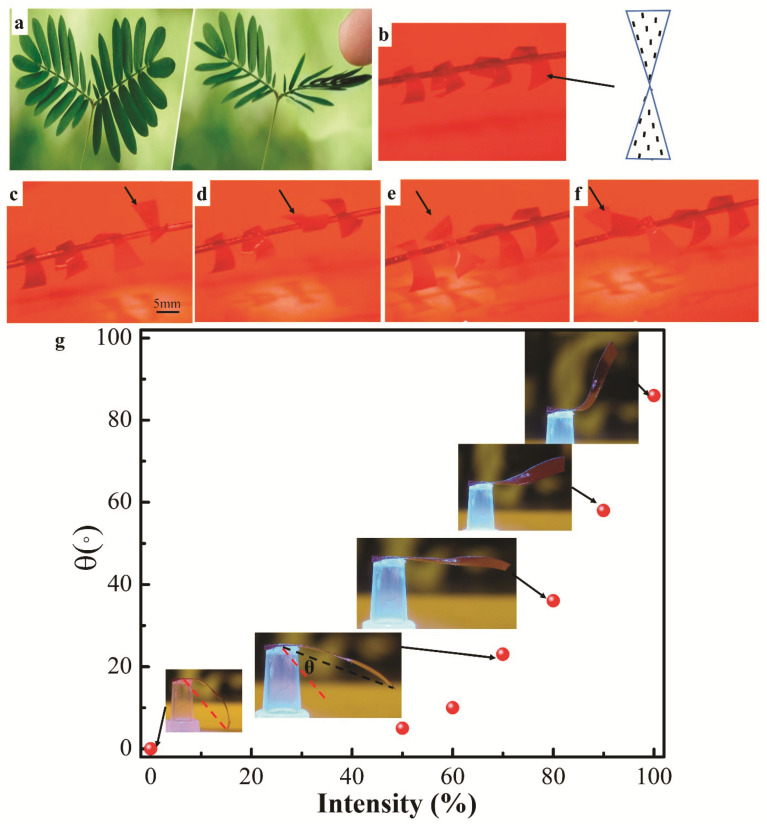
Bioinspired LCE *Mimosa Pudica*. (**a**) Leaves of plant *Mimosa Pudica* close immediately when stimulated by external forces. (**b**) The bioinspired LCE *Mimosa Pudica* before irradiation. The LCE assumes a hybrid configuration with the top surface pattern shown in the inset. (**c**–**f**) The “leaves” react to an external stimulus, “light”. The black arrow indicates the leaf pairs that are under irradiation. (**g**) The dependence of bending angle *θ* on light intensity.

## Data Availability

The data presented in this study are available on request from the corresponding author.

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
