# Peer review of "Light-Actuated Liquid Crystal Elastomer Prepared by Projection Display"

_materials, 2021, doi:10.3390/ma14237245_

Round 1

Reviewer 1 Report

The manuscript describes a new method to achieve photoalignment of liquid crystals for the preparation of patterned liquid crystal polymer films. Instead of a photomask or direct writing technique a projector is used to locally irradiate the photoalignment layer. I find the design of this technique very clever and creative, and I support the publication of the paper if a few small issues are addressed:

-In figures 2 and 3, the caption mentions a "red plate" is used in the optical analysis of the LC films. It is not clear what this is or what purpose it serves.

-Why are the alignment cells used a combination of a homeotropic and planar alignment layer? Why not make a cell with two photoalignment layers and then photoalign the whole cell at once?

-At what wavelengths does the projector emit light, and how does this relate to the absorbance of the photoalignment material?

Reviewer 2 Report

In this work, a photopatterning procedure based on a maskless projection display is presented. The method allows for the creation of a spatially-controlled director field in a liquid crystal elastomer (LCE). The main advantage of the method is its simplicity with respect to other photopatterning procedures based on masks or waveplates. The patterned LCEs are used as alignment surfaces in hybrid cells that  are filled with a  photopolimerizable LC mixture. The obtained polymer film is photoresponsive and its shape can be reversibly modified by irradiation of 455nm light.

The work is interesting and presents an innovative recipe for photopatterning. The description of the experiments and the interpretation of the results are clear and quite convincing. The manuscript is well written. Therefore, I think that the work is suitable for publication in Materials. Only some minor points should be addressed.

- In the work, the spectral characteristics of the  projector source are not described. This information should be included in the manuscript.

-No comment is made about the time stability of the photopatterned surfaces. For example, are they affected by day light?

-A brief comment about the spatial resolution of the patterning method would be desired, i.e., what is the smallest possible size for the topological defects ?

Reviewer 3 Report

This manuscript provides a purely phenomenological description of both the method and the physical phenomena involved. For this to call a real research paper I would expect a more in-depth analysis of e.g. the attainable resolution of the photopatterning method on one hand, and deformation of the elastomer film on the other. The authors present a cheap and clever photopatterning technique which, however, seems to be a very limited and rather low-resolution method of producing continuously varying director fields.  Comparing it to high-resolution, very precise lithography or AFM rubbing approaches seems unfair. There do, however, exist photopatterning techniques, both flexible and high-precision such as plasmonic photopatterning (see e.g. https://doi.org/10.1002/adma.201506002). Did the authors deliberately ignore it? Please, make an appropriate remark in the introduction and in the manuscript's main body. 

I realize an extensive theoretical approach is out of the scope of this paper but why not enrich the discussion with a more in-depth analysis of what really happens in the elastomer. For an example of what I mean see e.g.  https://doi.org/10.1098/rspa.2021.0259.

In conclusion, this manuscript reads well, looks interesting, and deserves publication, but requires a little bit of effort to turn a "press release" into a research paper.

Reviewer 4 Report

Paper presents a simple photopatterning method (based on a maskless projection display system) to create spatially varying molecular orientation in LCE films. I found it interesting; however, I am not so sure if it is interesting enough for a wide audience.

Some minor corrections/modifications are required:

  • line 36: "In order to fabricate LC (...)" - please note that LC is not fabricated
  • line 50: is should be "linear polarization direction"
  • line 51: "(...) used to control the orientation of LC molecules" - it is worth to add how the molecules are aligned with respect to the dye molecules
  • line 72: what the authors mean by "(...) with origami or kirigami designs"? 
  • line 85: it is worth to add letters to indicate the chemical structures of specific substances
  • line 86, please precize the temperature in which the substrate has been baked
  • line 87: please specified the plane in which molecules of azo-dye are oriented
  • line 89: please add the letters to indicate which chemical structures corresponds to which substance
  • line 91: usage of projection display for photopolymerization is not new - I suggest to remove "newly"
  • line 98: is an SD1 substrate placed on the screen - maybe the plane when the substrate is irradiated could be named in different way
  • line 114: maybe it is better to say that the polarizer axis is aligned along y-axis at the begining of the process that saying "the polarizer starts along the y-axis" - it is too big simplification for me
  • line 142: as previously stated - it is better to add letters to Fig. 1 to indicate different substances
  • line 205: "self-weight" - it would be usefull to know what was the weight of the "leave".
  • Results in Fig. 5: it would be interesting to describe what happend when illumination is off. Does the "leaf" go back to its initial position? How long does it take? Are the angles presented in Fig. 5g measured individually in each case of irradiation? What happens if we start from one (no zero position) and increase the light intensity? What is the light intensity/beam density corresponding to the intensity of 100%? Was the light beam focused on the "leaf"? What was the beam size/shape?

Even if the experimental part is relatively simple (with low scientific sound) I recommend paper publication as the proposed solution may be further used in soft robotics or biomedical devices.
